# Tuning the Hyperparameters of Anytime Planning: A Deep Reinforcement Learning Approach

**Abhinav Bhatia, Justin Svegliato, Shlomo Zilberstein**

College of Information and Computer Sciences
University of Massachusetts Amherst
{abhinavbhati, jsvegliato, shlomo}@cs.umass.edu

## Abstract

Many anytime algorithms have adjustable hyperparameters that affect their speed and accuracy. However, while existing work on metareasoning has focused on deciding when to interrupt an anytime algorithm and act on the current solution, there has not been much work on tuning the hyperparameters of an anytime algorithm at runtime. This paper introduces a decision-theoretic metareasoning approach that can optimize both the hyperparameters and the stopping point of *adjustable algorithms* with deep reinforcement learning. First, we propose a generalization of an anytime algorithm called an adjustable algorithm that has hyperparameters that can be tuned at runtime. Next, we offer a meta-level control technique that monitors and controls an adjustable algorithm by using deep reinforcement learning. Finally, we demonstrate that an application of our approach to anytime weighted A* is effective on a range of common benchmark problems.

## Introduction

Anytime algorithms often have *hyperparameters* that can be tuned to optimize their performance in a specific scenario—given a certain problem instance and time constraint. Simply put, an anytime algorithm is an algorithm that gradually improves its solution at runtime and can be interrupted for its solution at any time (Zilberstein 1996). This offers a trade-off between the quality and computation time of a solution that has proven to be useful in many real-time decision-making problems, such as motion planning (Karaman et al. 2011), heuristic search (Burns, Ruml, and Do 2013; Cserna, Ruml, and Frank 2017), object detection (Karayev, Fritz, and Darrell 2014), belief space planning (Pineau, Gordon, and Thrun 2003; Spaan and Vlassis 2005), and probabilistic inference (Ramos and Cozman 2005). Existing work on metareasoning that manages this trade-off focuses on deciding when to interrupt an anytime algorithm and act on the current solution. However, the scope of metareasoning can be extended to tune the hyperparameters of an anytime algorithm at runtime in order to optimize its performance.

There has been substantial work on developing metareasoning techniques for determining when to interrupt an anytime algorithm and act on the current solution. Generally, these techniques monitor and control an anytime algorithm in order to track its performance and calculate its stopping point at runtime. For example, an early technique models optimal stopping as a sequential decision problem and derives a meta-level control policy using dynamic programming (Hansen and Zilberstein 2001) while a more recent technique estimates the optimal stopping point using online performance prediction (Svegliato, Wray, and Zilberstein 2018). These techniques, however, cannot tune the hyperparameters of the anytime algorithm at runtime.

Metareasoning techniques for tuning the hyperparameters of an anytime algorithm at runtime have typically been designed for specific anytime algorithms. For instance, there has been work on tuning the weight of an anytime heuristic search algorithm called *anytime weighted A** by selecting the best weight for a problem (Hansen and Zhou 2007), choosing the best weight for an instance of a problem (Sun, Druzdzel, and Yuan 2007), and modifying the weight heuristically (Thayer and Ruml 2009). There has also been work on tuning the growth factor of an anytime motion planning algorithm called *RRT** (Urmson and Simmons 2003). Nevertheless, these techniques often have several drawbacks because they cannot determine the optimal stopping point, require expertise in the implementation of the anytime algorithm, and lack generality or formal analysis.

We therefore propose a decision-theoretic metareasoning approach for optimal stopping and hyperparameter tuning of *adjustable algorithms*. This approach models monitoring and controlling adjustable algorithms as a *Markov decision process* with four main attributes. The *states* reflect the quality and computation time of the current solution and any other features needed to summarize the internal state of the adjustable algorithm, the instance of the problem, and the performance of the system. The *actions* reflect either interrupting or executing the adjustable algorithm for another time step while tuning its internal hyperparameters. However, while the *reward function* can be determined easily, the *transition function* is unknown given the considerable complexity and uncertainty over the states and actions. In response, we use deep reinforcement learning, which is a model-free approach, to learn when to stop and how to tune the hyperparameters of the anytime algorithm online by learning through a series of episodes that each use the adjustable algorithm to solve an instance of a problem on the system. Generally, reinforcement learning is a robust approach to metareasoning for adjustable algorithms given the abundance of data that can be generated through simulation.

Our primary contributions in this paper are: (1) a generalization of an anytime algorithm called an adjustable algorithm, (2) a meta-level control technique that monitors and controls an adjustable algorithm based on deep reinforcement learning, (3) an application of our approach to anytime weighted A*, and (4) a demonstration that the application of our approach to anytime weighted A* is effective on a range of common benchmark problems.

## Related Work

Approaches to automatic hyperparameter tuning for general algorithms fall into two main categories. On the one hand, *model-based* approaches typically interleave fitting a model with selecting hyperparameters based on that model. Notably, building on earlier work in sequential model-based optimization (Bartz-Beielstein, Lasarczyk, and Preuss 2005; Hutter et al. 2009a, 2010), *SMAC* selects the hyperparameters of an algorithm using a model represented as a random forest (Hutter, Hoos, and Leyton-Brown 2011). On the other hand, while *model-free* approaches do not fit any kind of model, they have still been shown to be effective across a range of domains. Limited to numerical hyperparameters, *CALIBRA* uses experimental designs to find initial hyperparameters followed by local search to improve those hyperparameters (Adenso-Diaz and Laguna 2006) while *F-Race* leverages racing algorithms from machine learning (Birattari et al. 2002, 2010) to select the hyperparameters of an algorithm. Extending this work to categorical hyperparameters, *GGA* employs parallel gender-based genetic algorithms (Ansótegui, Sellmann, and Tierney 2009) while *ParamILS* performs iterated local search (Hutter, Hoos, and Stützle 2007; Hutter et al. 2009b) to select the hyperparameters of an algorithm.

However, while all of these methods have mostly been designed for general algorithms, they often do not take advantage of the benefits of anytime algorithms. By using deep reinforcement learning to monitor and control anytime algorithms in particular, our metareasoning approach avoids many drawbacks of earlier work. First, while some existing methods are limited to numerical hyperparameters and deterministic algorithms, our approach can support both categorical hyperparameters and stochastic algorithms as well. Next, unlike existing methods that only optimize the hyperparameters for a *single* problem instance, our approach can optimize the hyperparameters for *multiple* problem instances. In fact, our approach can not only select the hyperparameters of an anytime algorithm for a specific problem instance but also adjusts the hyperparameters of an anytime algorithm as it runs. Finally, though some existing methods cannot terminate a trial early, our approach can naturally terminate trials early by using deep reinforcement learning.

There has also been an orthogonal line of research on using a portfolio of algorithms to solve different instances of hard computational problems. In fact, it has been recognized that different algorithms tend to dominate each other on different instances of a problem because there is often no single best algorithm (Leyton-Brown et al. 2003). This has resulted in methods that can use *portfolios of algorithms* for

satisfiability (Gomes and Selman 2001; Xu et al. 2008), *ensemble methods* in machine learning (Dietterich 2000; Fern and Givan 2003), and *multiple methods* in real-time problem solving (Wagner, Garvey, and Lesser 1998; Zilberstein and Mouaddib 2000). More recently, SATzilla (Xu et al. 2008), an efficient solver that uses a portfolio of algorithms to solve difficult satisfaction problems, has won multiple competitions and continues to dominate the field.

Optimal stopping for anytime algorithms but not hyperparameter tuning has been well-studied as well. The earliest approach, *fixed allocation*, executes the anytime algorithm until a stopping point determined prior to runtime (Horvitz 1987; Boddy and Dean 1994). While fixed allocation is effective given negligible uncertainty in the performance of the anytime algorithm, there is often considerable uncertainty in real-time planning problems (Paul et al. 1991). In response, a more sophisticated approach, *monitoring and control*, tracks the performance of the anytime algorithm and estimates a stopping point at runtime periodically (Horvitz 1990; Zilberstein and Russell 1995; Hansen and Zilberstein 2001; Lin et al. 2015; Svegliato, Wray, and Zilberstein 2018; Svegliato, Sharma, and Zilberstein 2020). We note that our approach not only determines the stopping point of an anytime algorithm but also tunes its hyperparameters.

With respect to heuristic search, there is some recent work on dynamically selecting the heuristic function using deep reinforcement learning (Speck et al. 2021) and dynamically adjusting the broader search strategy in classical planning using evolutionary strategies (Gomoluch et al. 2020). However, our work is not specific to heuristic search and focuses on anytime algorithms.

## Meta-Level Control Problem

We begin by reviewing the standard meta-level control problem for anytime algorithms. This requires a function that describes the utility of a solution computed by an anytime algorithm in terms of its quality and computation time (Horvitz and Rutledge 1991). We define this function below.

**Definition 1.** *A **time-dependent utility function** $U : \Phi \times \Psi \to \mathbb{R}$ represents the utility $U(q, t)$ of a solution of quality $q \in \Phi$ at time step $t \in \Psi$.*

A time-dependent utility function can often be expressed as the difference between an *intrinsic value function* and a *cost of time* (Horvitz 1988; Hansen and Zilberstein 2001). An intrinsic value function represents the utility of a solution given its quality but not its computation time while a cost of time represents the utility of a solution given its computation time but not its quality. We define this property below.

**Definition 2.** *A time-dependent utility function $U : \Phi \times \Psi \to \mathbb{R}$ is **time-separable** if the utility $U(q, t)$ of a solution of quality $q \in \Phi$ at time step $t \in \Psi$ can be expressed as the difference between two functions $U(q, t) = U_I(q) - U_C(t)$ where $U_I : \Phi \to \mathbb{R}^+$ is the **intrinsic value function** and $U_C : \Psi \to \mathbb{R}^+$ is the **cost of time**.*

The standard meta-level control problem for anytime algorithms is the problem in which an intelligent system must

determine the point at which to interrupt an anytime algorithm and act on the current solution (Horvitz 1990; Zilberstein 1996). The anytime algorithm should ideally be interrupted at the optimal stopping point since it is the optimum of the time-dependent utility function. However, the optimal stopping point is often challenging to determine due to substantial uncertainty over the performance of the anytime algorithm and urgency for the solution.

## Adjustable Algorithms

We propose an *adjustable algorithm*, a generalization of an anytime algorithm, with *internal state* that can be *monitored* and *internal hyperparameters* that can be *controlled* to either interrupt or execute the algorithm for another time step and adjust the internal operation of the algorithm. This results in a new meta-level control problem that involves optimal stopping and optimal hyperparameter tuning of the adjustable algorithm. Generally, our approach monitors and controls an adjustable algorithm by expressing this novel meta-level control problem as a deep reinforcement learning problem. Hence, an adjustable algorithm is defined as follows.

**Definition 3.** *An **adjustable algorithm**, $\Lambda$, has internal state that can be monitored and internal hyperparameters $\{\Theta_0, \Theta_1, \ldots, \Theta_{N_\Theta}\}$ that can be controlled such that the internal hyperparameter $\Theta_0 = \{\text{STOP}, \text{CONTINUE}\}$ either interrupts or executes the algorithm for another time step and the internal hyperparameters $\{\Theta_1, \ldots, \Theta_{N_\Theta}\}$ adjust the internal operation of the algorithm.*

*Anytime weighted A\**, a popular anytime algorithm that extends the well-known A\* heuristic search algorithm (Hansen, Zilberstein, and Danilchenko 1997; Likhachev, Gordon, and Thrun 2004; Aine, Chakrabarti, and Kumar 2007; Hansen and Zhou 2007; Thayer and Ruml 2010), is an example of an adjustable algorithm. We describe anytime weighted A\* in detail to illustrate our approach. Anytime weighted A\* (1) uses an inadmissible heuristic to quickly find suboptimal solutions, (2) continues the search after each solution is found, (3) provides an error bound on each suboptimal solution, and (4) guarantees an optimal solution once terminated. Notably, the standard evaluation function $f(n) = g(n) + h(n)$ used to select the next node for expansion from the open list is replaced with a weighted evaluation function $f_w(n) = g(n) + w \cdot h(n)$, where the path cost $g(n)$ is the cost of the path from the start node to a node $n$ and the heuristic $h(n)$ is the estimated cost from a node $n$ to the goal node. Given a weight $w>1$, the weighted heuristic becomes potentially inadmissible and the algorithm prioritizes expanding a node that appears closer to a goal node by weighting the heuristic component $h(n)$ more heavily than the path cost component $g(n)$. This causes the algorithm to speed up computation time at the expense of solution quality, which is inversely proportional to cost.

Figure 1 shows typical performance curves for two executions of anytime weighted A\* that solve an instance of a problem with different weights. This figure illustrates two cases. With *deadlines*, a weight of 2.0 leads to better quality at *Contract 1* while a weight of 1.5 results in better quality at *Contract 2*. Without *deadlines*, a weight of 2.0 leads

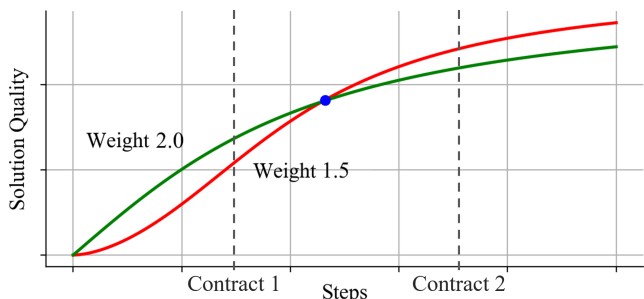

Figure 1: An example of two executions of anytime A\*.

to better quality in the short term but worse quality in the long term while a weight of 1.5 results in worse quality in the short term but better quality in the long term. This raises two important question: (1) How can metareasoning dynamically tune the hyperparameters of an adjustable algorithm at runtime in order to optimize the trade-off between solution quality and computation time *with* or *without* deadlines? (2) Can a general approach be developed to learn to perform this optimization automatically? We answer these questions by offering a disciplined general-purpose metareasoning approach to optimal stopping and optimal hyperparameter tuning of adjustable algorithms in this paper.

Our approach to monitoring and controlling adjustable algorithms uses deep reinforcement learning to express the meta-level control problem as a *Markov decision process* (MDP) (Bellman 1966). An MDP is a formal decision-making model for reasoning in fully observable, stochastic environments that can be defined by a tuple $\langle S, A, T, R, s_0 \rangle$, where $S$ is a finite set of states, $A$ is a finite set of actions, $T : S \times A \times S \rightarrow [0, 1]$ represents the probability of reaching a state $s' \in S$ after performing an action $a \in A$ in a state $s \in S$, $R : S \times A \times S \rightarrow \mathbb{R}$ represents the expected immediate reward of reaching a state $s' \in S$ after performing an action $a \in A$ in a state $s \in S$, and $s_0 \in S$ is a start state. A solution to an MDP is a policy $\pi : S \rightarrow A$ indicating that an action $\pi(s) \in A$ should be performed in a state $s \in S$. A policy $\pi$ induces a value function $V^\pi : S \rightarrow \mathbb{R}$ representing the expected discounted cumulative reward $V^\pi(s) \in \mathbb{R}$ for each state $s \in S$ given a discount factor $0 \leq \gamma < 1$. An optimal policy $\pi^*$ maximizes the expected discounted cumulative reward for every state $s \in S$ by satisfying the Bellman optimality equation $V^*(s) = \max_{a \in A} \sum_{s' \in S} T(s, a, s')[R(s, a, s') + \gamma V^*(s')]$.

We express the meta-level control problem for monitoring and controlling adjustable algorithms as an MDP with two primary attributes. First, the set of *states* has *state factors* that reflect the quality and computation time of the current solution but can also have *state factors* that represent the internal state of the algorithm, the instance of the problem, or the performance of the system. Second, the set of *actions* has an *action factor* that reflects the internal hyperparameter that either interrupts or executes the algorithm for another time step but can also have *action factors* that represent the internal hyperparameters that adjust the internal operation of the algorithm. We present a description of this MDP below.

**Definition 4.** *The **meta-level control problem** for monitoring and controlling an adjustable algorithm, $\Lambda$, is represented by an MDP $\langle \Phi, \Psi, F, S, A, T, R, s_0 \rangle$ given a time-dependent utility function $U : \Phi \times \Psi \to \mathbb{R}$:*

- $\Phi = \{q_0, q_1, \ldots, q_{N_\Phi}\}$ *is a set of qualities.*

- $\Psi = \{t_0, t_1, \ldots, t_{N_\Psi}\}$ *is a set of time steps.*

- $F = F_0 \times F_1 \times \cdots \times F_{N_F}$ *is a set of features that summarize the internal state of the algorithm, the instance of the problem, or the performance of the system.*

- $S = \Phi \times \Psi \times F$ *is a set of states of computation: each state $s \in S$ indicates that the algorithm has a solution of quality $q \in \Phi$ at time step $t \in \Psi$ with a feature $f \in F$.*

- $A = \Theta_0 \times \Theta_1 \times \cdots \times \Theta_{N_\Theta}$ *is a set of actions of computation: the internal hyperparameter $\Theta_0 = \{\text{STOP}, \text{CONTINUE}\}$ either interrupts or executes the algorithm for another time step while the internal hyperparameters $\Theta_1, \ldots, \Theta_{N_\Theta}$ adjust the internal operation of the algorithm.*

- $T : S \times A \times S \to [0, 1]$ *is an unknown (possibly nonstationary) transition function that represents the probability of reaching a state $s' = (q', t', f') \in S$ after performing an action $a \in A$ in a state $s = (q, t, f) \in S$.*

- $R : S \times A \times S \to \mathbb{R}$ *is a reward function that represents the expected immediate reward, $R(s, a, s') = U(q', t') - U(q, t)$, of reaching a state $s' = (q', t', f') \in S$ after performing an action $a \in A$ in a state $s = (q, t, f) \in S$.*

- $s_0 \in S$ *is a start state $s_0 = (q_0, t_0, f_0) \in S$ that indicates that the algorithm has a solution of quality $q_0 \in \Phi$ at time step $t_0 \in \Psi$ with a feature $f_0 \in F$.*

Note that the reward function is consistent with the objective of optimizing the time-dependent utility function: executing the adjustable algorithm until a solution of quality $q \in \Phi$ at time step $t \in \Psi$ with a feature $f \in F$ gives a cumulative reward that is equal to the time-dependent utility $U(q, t)$.

Many approaches to meta-level control of *anytime algorithms* use solution quality and computation time for the state of computation (Zilberstein and Russell 1995). Such a state of computation, however, may not satisfy the *Markov* property. It could therefore benefit from features that summarize the internal state of algorithm, the instance of the problem, or the performance of the system. For example, in a domain where anytime weighted A* solves an instance of a TSP, there could be features for the mean and standard deviation of the $g$- and $h$-values across the nodes on the open list of anytime weighted A*, the number of cities in the instance of the TSP (Hutter et al. 2014), or the processor usage or memory pressure of the system. Our metareasoning approach for *adjustable algorithms* can employ a more sophisticated and rich representation for the state of computation thanks to deep reinforcement learning.

We show that an optimal policy of the meta-level control problem produces optimal meta-level control of an adjustable algorithm under certain conditions below.

**Remark 1.** *If the change in the current solution of the adjustable algorithm given a state of computation $s \in S$ and*

---

**Algorithm 1** Our technique that uses DQN to learn the optimal stopping point and hyperparameters.

---

**Require:** An anytime algorithm $\Lambda$, an action-value network $\mathcal{N}$, a step size $\alpha_1$, a target action-value network step size $\alpha_2$, an exploration strategy $\mathcal{E}$, an experience buffer capacity $\ell_1$, a number of episodes $\ell_2$, an initialization period $\ell_3$, a minibatch size $\ell_4$, and a duration $\Delta$

1: $B \leftarrow \text{EXPERIENCEBUFFER}(\ell_1)$
2: $Q \leftarrow \text{NEURALNETWORK}(\mathcal{N})$
3: $\hat{Q} \leftarrow Q$
4: **for** $i = 1, 2, \ldots, \ell_2$ **do**
5:     $P \leftarrow \text{SAMPLEPROBLEMDISTRIBUTION}()$
6:     $\Lambda.\text{SETUP}(P)$
7:     $t \leftarrow 0$
8:     $s_t \leftarrow (q_0, t_0, \Lambda.\text{GET}F())$
9:     $a_t \leftarrow \pi_{\mathcal{E}}^Q(s_t)$
10:    $\Lambda.\text{START}(a_t.\Theta_1, \ldots, a_t.\Theta_{\ell_\Theta})$
11:    $\text{SLEEP}(\Delta)$
12:    **while** $\Lambda.\text{RUNNING}()$ **do**
13:        $t' \leftarrow t + \Delta$
14:        $s_{t'} \leftarrow (\Lambda.\text{GET}\Phi(), \Lambda.\text{GET}\Psi(), \Lambda.\text{GET}F())$
15:        $r_t \leftarrow R(s_t, a_t, s_{t'})$
16:        $B.\text{APPEND}((s_t, a_t, r_t, s_{t'}))$
17:        **if** $B.\text{SIZE}() \geq \ell_3$ **then**
18:           $M \leftarrow B.\text{SAMPLEMINIBATCH}(\ell_4)$
19:           $\hat{\mathcal{L}}(r, s') := r + \gamma \max_{a' \in A} \hat{Q}(s', a')$
20:           $\mathcal{L}(s, a, r, s') := [\hat{\mathcal{L}}(r, s') - Q(s, a)]^2$
21:           $Q.\text{BACKPROPAGATE}(M, \mathcal{L}, \alpha_1)$
22:           $\hat{Q} \leftarrow (1 - \alpha_2) \cdot \hat{Q} + \alpha_2 \cdot Q$
23:        $t \leftarrow t'$
24:        $a_t \leftarrow \pi_{\mathcal{E}}^Q(s_t)$
25:        **if** $a_t.\Theta_0 = \text{STOP}$ **then**
26:           $\Lambda.\text{STOP}()$
27:           **break**
28:        $\Lambda.\text{CONTINUE}(a_t.\Theta_1, \ldots, a_t.\Theta_{\ell_\Theta})$
29:        $\text{SLEEP}(\Delta)$
30:    **return** $Q$

---

*an action of computation $a \in A$ satisfies the Markov property, the optimal policy $\pi^* : S \to A$ results in optimal stopping and optimal hyperparameter tuning.*

*Proof Sketch.* This follows directly from the Markov property: a transition to a successor state of computation $s' \in S$ only depends on the current state of computation $s \in S$ and the current action of computation $a \in A$. Note that while using only quality and time for the state of computation is unlikely to satisfy the Markov property, this richer representation allows for a better approximation. $\square$

## Meta-Level Control Technique

Our meta-level control technique monitors and controls adjustable algorithms by performing episodes that each use the adjustable algorithm to solve an instance of a specific problem on a given system with deep reinforcement learning.

Deep reinforcement learning has been effective across a wide range of applications, including Atari (Mnih et al. 2015), chess (Silver et al. 2018), and StarCraft (Vinyals et al. 2019). A deep reinforcement learning agent learns a

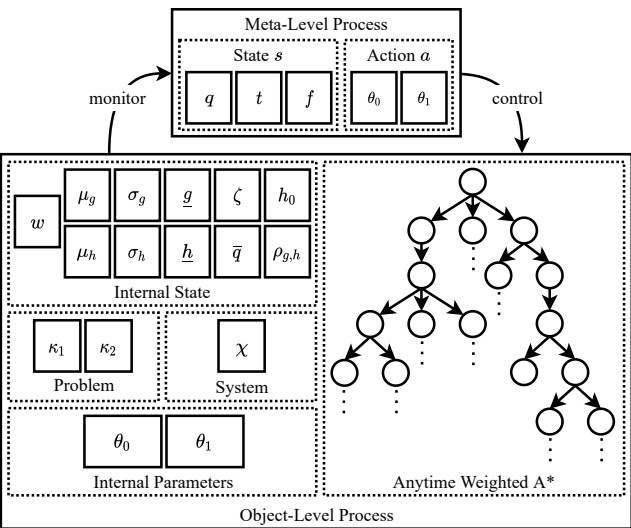

Figure 2: The metareasoning architecture that has a meta-level process and an object-level process.

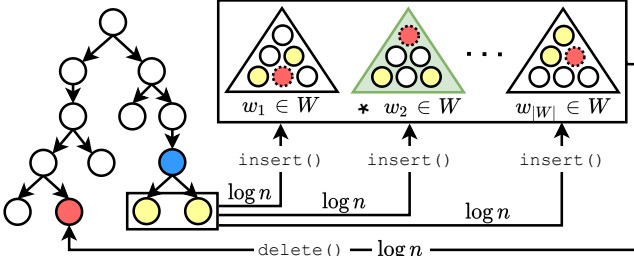

Figure 3: A modified implementation of anytime weighted A* that maintains multiple open lists.

value function or a policy expressed as a neural network by taking actions and observing rewards in its environment. This makes it a natural approach to adjustable algorithms for three reasons (Sutton and Barto 1998). First, by balancing exploitation with exploration, it can learn a policy that tunes the internal hyperparameters of an adjustable algorithm without knowing the transition function. Next, by ignoring large regions of the state space not reached in practice, it can reduce the overhead of learning a policy that tunes the internal hyperparameters of an adjustable algorithm. Finally, by using a neural network that can learn complex patterns between massive input and output spaces, it can learn the effect of the internal hyperparameters on the internal state of an adjustable algorithm.

Algorithm 1 shows our metareasoning technique for monitoring and controlling adjustable algorithms using DQN, *deep Q-learning* (Mnih et al. 2015), as described below.

**Reinforcement Learning Episode Loop**  The experience buffer is initialized to a capacity (Line 1). The current action-value function is initialized to the action-value network and the target action-value function is initialized to the current action-value function (Lines 2-3). A loop iterates over each episode (Line 4). For each episode, the following phases are performed for setup (Lines 5-11), monitor (Lines 12-15), update (Lines 16-22), and control (Lines 23-29). The action-value function is returned (Line 30).

**Episode Setup Phase**  An instance of the problem is sampled from the problem distribution and the anytime algorithm is setup to solve the instance (Line 5-6). The current time is initialized to zero (Line 7). The current state is initialized to the quality and computation time of the initial solution along with any extra features while the current action is initialized to the policy calculated from the current action-value function and an exploration strategy at the current state (Lines 8-9). The algorithm starts to solve the problem in-

stance with hyperparameters set according to the initial action (Lines 10-11).

**Episode Monitor Phase**  A loop runs until the algorithm is interrupted early or terminated naturally (Line 12). The successor state is set to the quality and computation time of the new solution and any extra features while the reward is calculated from the current state, the current action, and the successor state (Line 13-15). The experience buffer is appended with the current state, the current action, the reward, and the successor state (Line 16).

**Episode Update Phase**  The phase occurs if the size of the experience buffer is greater than an initialization period (Line 17). A minibatch is sampled from the experience buffer (Line 18). The loss function is defined as the square of the temporal-difference error (Line 19-20). The current action-value function represented as a neural network is updated via backpropagation from the minibatch and the loss function (Lines 21). The target action-value function is set to the current action-value function (Lines 22).

**Episode Control Phase**  The current time is incremented and the new current action is set to the policy calculated from the current action-value function and an exploration strategy at the new current state (Line 23-24). If the action indicates to stop the algorithm, the algorithm is interrupted and the loop goes to the next episode (Lines 25-27). Otherwise, the algorithm execution continues after tuning its internal hyperparameters (Line 28-29).

## Anytime Weighted A*

We now apply our approach to anytime weighted A*. Recent work on the algorithm focuses on selecting the best static weight for a problem (Hansen and Zhou 2007), choosing the weight for an instance of a problem (Sun, Druzdzel, and Yuan 2007), and modifying the weight at runtime heuristically (Thayer and Ruml 2009). The algorithm can also be improved via restarting when a solution is found (Richter, Thayer, and Ruml 2010). There has even been work that analyzes the failure conditions of the algorithm (Wilt and Ruml 2012). Overall, tuning the weight of anytime weighted A* at runtime has consistently proven to be challenging.

The meta-level control problem for anytime weighted A*, $\Lambda$, is an MDP $\langle \Phi, \Psi, F, S, A, T, R, s_0 \rangle$. $\Phi = [0, 1]$ is a set of qualities of the current solution. $\Psi = [0, \tau]$ is a set of time

steps with a deadline $\tau$. $F = W \times \mathbb{R}^9 \times [-1, 1] \times K \times [0, 1]$ is a set of features such that the feature $w \in W$ is the current weight, the features $\mu_g \in \mathbb{R}$ and $\mu_h \in \mathbb{R}$ are the mean of the $g$- and $h$-values on the open list, the features $\sigma_g \in \mathbb{R}$ and $\sigma_h \in \mathbb{R}$ are the standard deviation of the $g$- and $h$-values on the open list, the features $\underline{g} \in \mathbb{R}$ and $\underline{h} \in \mathbb{R}$ are the minimum $g$- and $h$-values on the open list, the feature $\zeta \in \mathbb{R}$ is the value $\log(n)$ for the number of nodes $n$ on the open list, the feature $\bar{q} \in \mathbb{R}$ is the $h$-value of the initial state divided by the minimum $f$-value on the open list, the feature $h_0$ is the $h$-value of the initial state, the feature $\rho_{g,h} \in [-1, 1]$ is the correlation between the $g$- and $h$-values on the open list, the feature $\kappa \in K$ is a set of settings for the instance of the problem, and the feature $\chi \in [0, 1]$ is the processor usage of the system. $A = \Theta_0 \times \Theta_1$ is a set of actions such that the internal hyperparameter, $\Theta_0 \in \{\text{STOP}, \text{CONTINUE}\}$, interrupts or executes the algorithm for another time step and the internal hyperparameter, $\Theta_1 = \{\oplus, \ominus\}$, tunes the weight $w \in W$ by a value $\nu \in \{-1, -\frac{1}{4}, \frac{1}{4}, 1\}$ within the bounds $1 \leq w \leq 5$. Note that $S$, $A$, $T$, $R$, and $s_0$ follow from the meta-level control problem for adjustable algorithms.

The metareasoning architecture for anytime weighted A* in Figure 2 has a *meta-level process* that monitors and controls an *object-level process* that executes the algorithm.

Anytime weighted A* involves a simple modification to allow its weight to be adjusted at runtime. Instead of inserting/deleting a node into/from a single open list for a static weight, the algorithm inserts/deletes this node into/from $|W|$ open lists each ordered by the $f_w$-value of a weight $w \in \mathcal{W}$ as illustrated in Figure 3. Consequently, each open list has a different ordering of the same set of nodes. Note that the time complexity of inserting/deleting a node across all open lists of size $n$ sequentially is $\mathcal{O}(|W| \log n)$.

## Experiments

In this section, we compare our approach for monitoring and controlling anytime weighted A* to a standard approach that uses a range of static weights of 1, 1.5, 2, 3, 4, and 5.

**Experimental Setup**    Each approach is evaluated using the mean *solution quality* on an identical set of 500 random instances after being trained for 12000 episodes across each benchmark problem. Note that training uses randomization seeds that are different than the randomization seeds used for evaluation and takes approximately one day for each benchmark problem on our system. Intuitively, a solution quality $q = 0$ represents no solution while a solution quality $q = 1$ represents an optimal solution. Formally, we define solution quality as the approximation ratio, $q = c^*/c$, where $c^*$ and $c$ is the cost of the optimal solution and the current solution. However, since it is infeasible to calculate the cost of the optimal solution, we estimate solution quality as the approximation ratio, $q = h(s_0)/c$, where $h(s_0)$ and $c$ is the heuristic value of the initial state and the cost of the current solution, like earlier work (Hansen and Zilberstein 2001).

A meta-level control problem has a time-dependent utility function. However, while it is possible to use any time-dependent utility function, we consider a *contract* setting in which the adjustable algorithm must terminate before a du-

ration of $\tau$ sec to avoid a significant utility penalty $\Upsilon$. This is common in robotics where a system has a fixed duration for planning before execution. Formally, given a solution of quality $q \in \Phi$ at time step $t \in \Psi$, the time-dependent utility function is $U(q, t) = [t \leq \tau] \cdot U_I(q) - [t > \tau] \cdot \Upsilon$, where $U_I(q) = \iota q$ is the utility of a solution of quality $q \in \Phi$ for an intrinsic value multiplier $\iota$.

Our meta-level control technique in Algorithm 1 has the hyperparameters below. The action-value network $\mathcal{N}$ is a fully connected neural network with two hidden layers of 64 and 32 nodes with ReLU activation and a linear output layer of 5 nodes. The last 5 observations are stacked and provided to the input layer of the neural network to observe 1 sec of runtime. The step size $\alpha_1$ is 0.0001. The target action-value network step size $\alpha_2$ is 0.001. The exploration strategy $\mathcal{E}$ is $\epsilon$-greedy action selection with an exploration probability $\epsilon$ that is annealed from 1 to 0.1 over 1000 episodes. The experience buffer capacity $\ell_1$ is $\infty$. The number of episodes $\ell_2$ is 12000. The initialization period $\ell_3$ is 10000. The minibatch size $\ell_4$ is 64. The duration $\Delta$ is 0.2 sec.

Each approach must solve benchmark problems that reflect domains that require different static weights and for which recent work reported counterintuitive behavior of anytime weighted A* (Wilt and Ruml 2012). Their settings are chosen to avoid trivializing the problem by either allowing too little time so that no approach finds a solution or too much time so that every approach finds the optimal solution.

***Sparse Traveling Salesman Problem***    An STSP instance has a set of $J$ cities that must be visited along an optimal route given a cost for each edge between a pair of cities where a percentage of edges have an infinite cost. The total distance of a minimum spanning tree across the unvisited cities is used as an admissible and consistent heuristic function. The number of cities $J$ is chosen randomly between 15 and 25. The percentage of edges with an infinite cost is chosen randomly between 0% and 30%. The cost between each pair of cities is chosen randomly. The MDP has a set of features $K = \kappa_1 \times \kappa_2$ that represent the number of cities $\kappa_1 \in \mathbb{N}^+$ and the percentage of edges with an infinite cost $\kappa_2 \in [0, 1]$.

***City Navigation Problem***    A CNP instance simulates navigating between two locations in different cities (Wilt and Ruml 2012). There is a set of $J$ cities scattered randomly on a $j \times j$ square such that each city is connected by a random tour and to a set of its nearest $n_J$ cities. Each city contains a set of $I$ locations scattered randomly throughout the city that is an $i \times i$ square such that each location in the city is connected by a random tour and to its nearest $n_I$ locations. The edge between a pair of cities costs the Euclidean distance plus an offset $\beta_1$. The edge between a pair of locations within a city costs the Euclidean distance scaled by a random number sampled between 1 and a maximum $\beta_2$. The goal is to find an optimal path from a randomly selected location in one city to randomly selected location in another city. The Euclidean distance from the current location to the target location is used as an admissible and consistent heuristic. The settings are chosen such that $J = 150$, $j = 100$, $n_J = 3$, $I = 150$, $i = 1$, $n_I = 3$, $\alpha = 2$, and $\beta = 1.1$. The MDP does

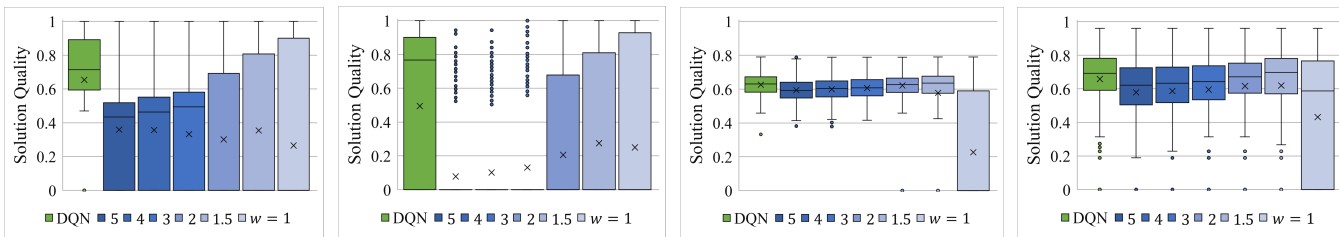

Figure 4: The box and whisker plots for the final solution qualities of each approach over all instances of SP, ISP, STSP, and CNP (*left* to *right*) The *crosses* denote the mean and the *bullets* denote the outliers. The median and the upper quartiles are even zero for some approaches.

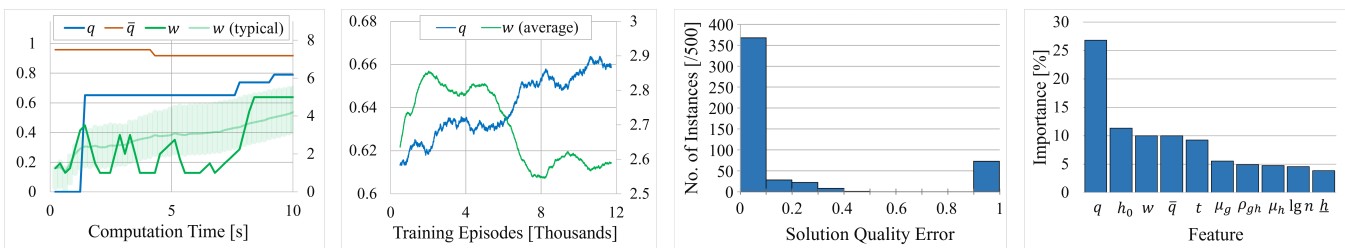

Figure 5: The analysis charts for SP from *left* to *right*: (a) A line chart showing how the solution quality, solution quality upper bound, and weight (plotted on secondary y-axis) change with computation time for a select instance. The faded line and shaded region indicate the mean weight and its standard deviation over all instances. (b) A line chart showing how the final solution quality and mean weight (on secondary y-axis) of an episode change with training episodes. (c) A histogram showing the distribution of all instances over solution quality error for our approach. *Solution quality error* is the normalized difference between the final solution quality of our approach and the final solution quality of the best approach. (d) A bar chart showing the percentage importance of the top 10 features that are learned by our approach. *Importance* is the mean absolute weight on a feature in the input layer of the neural network.

not include any instance specific features for this problem.

***Sliding Puzzle***   An SP instance has a set of $J = j^2 - 1$ tiles with each tile $i$ labeled from $1$ to $J$ in a $j \times j$ grid that must be moved from an initial position to a desired position given a *unit* cost $c(i) = 1$ for moving a tile $i$. The sum of the Manhattan distances from the current position of each tile to a desired position is used as an admissible and consistent heuristic function. The number of tiles is 15. The difficulty of the initial position, as measured by the heuristic function is chosen randomly between $35$ and $45$, and included in the feature space of the MDP.

***Inverse Sliding Puzzle***   An ISP instance is the same as an SP instance except that there is an *inverse* cost $c(i) = 1/i$ for moving a tile $i$. This means that the sum of the Manhattan distances from the current position of each tile to the desired position, *weighted* by the cost for moving each tile, is used as an admissible and consistent heuristic function.

The duration $\tau$ is $10$ sec for SP and ISP, $5$ sec for STSP, and $4$ sec for CNP corresponding to $6000$, $3000$, and $2400$ node expansions on an AMD Ryzen 3900X processor with 32 GB of 3200 MHz memory. Each approach runs until the node expansion limit instead of the duration for experimental consistency and reproducibility.

**Experimental Results**   Our approach solves more problem instances and produces higher average solution quality compared to the standard approach with any static weight.

Figure 4 shows the results for SP, ISP, STSP, and CNP. Our approach beats the standard approach with the best static weight: it has a substantially higher mean solution quality in SP for $w = 4$ and ISP for $w = 1.5$, a comparable mean solution quality in STSP for $w = 2$, and fairly higher mean solution quality in CNP for $w = 2$.

Figure 5 shows four analyses for SP. In Figure 5(a), our approach adjusts the weight based on solution quality and the other features that are not listed. Generally, the mean weight increases as the computation time increases to ensure generating at least one solution. In Figure 5(b), our approach improves its final solution quality with each training episode by learning gradually. In fact, the average weight initially increases but then decreases to generate higher quality solutions. In Figure 5(c), our approach exhibits a solution quality error of 0 for over 350 instances. While roughly 50 instances have a solution quality error of 1, this is still better than the standard approach. Figure 5(d) shows that our approach takes advantage of the other features, such as the current weight, the upper bound on solution quality, and the initial heuristic value, in addition to solution quality and computation time.

Our open source library[1] provides an OpenAI Gym environment for monitoring and controlling adjustable algorithms using deep reinforcement learning.

---

[1]https://github.com/bhatiaabhinav/model-free-metareasoning

## Conclusion

This paper introduces a *nonmyopic*, *decision-theoretic*, and *general-purpose* metareasoning approach that optimizes the hyperparameters and stopping point of adjustable algorithms. First, we propose a generalization of an anytime algorithm called an adjustable algorithm. We then offer a meta-level control technique that monitors and controls an adjustable algorithm by using deep reinforcement learning. Finally, we apply our approach to anytime weighted A* and show that it is effective on a range of problems. Future work will explore how to apply our approach to other common task and motion planning algorithms, such as tuning the growth factor of the motion planning algorithm RRT*.

## Acknowledgments

This work was supported in part by the NSF Graduate Research Fellowship DGE-1451512, and NSF grants IIS-1813490 and IIS-1954782.

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
