# OpenReview forum: "Tuning the Hyperparameters of Anytime Planning: A Deep Reinforcement Learning Approach"
_icaps-conference.org/ICAPS/2021/Workshop/HSDIP — HSDIP 2021_

### Official Review · AnonReviewer2 · 2021-05-25
**The paper is well written and technically of good quality. The paper is weaker on the evaluation side, esp. not showing limitations of the proposed approach.**

**Confidence:** 4
**Overall Score:** Weak Accept

**Review:**

The paper is well written and organized and easy to read. Technically it is sound and contains all the theoretical details I was expecting. The idea of using Deep RL (DRL) for meta-parameter tuning of search is interesting and significant as it can help wide variety of search problems.

My biggest reservations are with the experimental setup. First the modification of the WA* algorithm does not seem to me as "simple" as described by the authors. It principally provides multi-weighted A* search, therefore the question is whether this change is not enough to provide similar results as with the whole DRL machinery on top of it. Even if not, this has to be theoretically (or maybe more easily experimentally) shown in the paper. The provided experiments compare the DRL approach with fixed weights of several WA*, but not the multi-weighted A* without the DLR part. Second, I am missing details on how the particular benchmark problems were reflected in the parametrization of the DLR-based approach. The paper mentions features for the Sparse Traveling Salesman Problem, but for the other benchmark problems, only the way they were generated is described but without details on how the DLR problem was mapped to them. Finally, the approach was not experimented cross problem to show how the DLR generalizes the learned knowledge between different problems and where are the limits of the proposed approach.

Minor comments:
* "The optimal stopping point is often challenging to determine, however, due to substantial uncertainty over the performance of the anytime algorithm or the urgency for the solution." -- I don't get the sentence
* Def 3. "internal state" would benefit formal definition
* Alg. 1, lines 15-21 would benefit a more detailed explanation then just "update"
* "feature $\kappa \in K$ is a set of settings for the instance of the problem" -- the settings should be concertized
* "Sp and Isp , 5 sec for TSP, and 4 sec for Cnp" -- forward references the benchmark problems which will be used

---

> ### Author Response · Authors · 2021-06-04
> **Author Response**
>
> Thank you for the useful feedback and helpful suggestions.
>
> Please note that multi-weighted A* is AWA* with the ability to adjust weights at runtime. Since it is not a standalone algorithm if an adjustment policy is not specified, it is not a baseline. However, given your concern, we will make our best effort to use multi-weighted A* as a baseline by developing a simple adjustment policy. We will clarify this.
>
> Thank you for noticing that we unintentionally omitted the state features for some benchmark domains. For SP and ISP, the metareasoner observes the Manhattan distance between the initial state and the goal puzzle. For CNP, the metareasoner does not observe any problem instance specific features. We will mention this in the paper.
>
> We agree that it would be useful to analyze the generalization performance of a trained metareasoner across different benchmark problem domains. We appreciate the idea, and we will perform such experiments in future work.
>
> We believe that the main weakness of our approach is that it requires offline training on large amounts of data. While we mention in our paper that a cheap and fast simulator is usually available, we admit that this is not always true. In such cases, our approach would not be as feasible. We will emphasize this.
>
> We will address all other minor comments in the paper.

---

### Official Review · AnonReviewer1 · 2021-05-26

**Confidence:** 3
**Overall Score:** Weak Accept

**Review:**

**Tittle: Metareasoning for tuning the Hyperparameters of Anytime Planning: A Deep Reinforcement Learning Approach**

### Summary
An anytime algorithm is an algorithm that improves the solution quality over time. It can be interrupted at any point in time and returns the best solution found so far. For a given problem, we can come up with a utility function which takes into account the time to find a solution and the solution quality. A lot of research has been done to estimate when the anytime algorithm should be stopped to optimize the utility.

The authors generalize anytime algorithms to adjustable algorithms. Adjustable algorithms can change their hyperparameters during execution. They model the control problem as an MDP and use RL to learn a controller that adjust the parameters and decides when to stop. This is then  implemented for anytime weighted A* and evaluated on *Sparse Traveling Salesman Problem*, *City Navigation Problem*, *Inverse sliding Puzzle*.


### Feedback
The paper is readable, some sections can be improved. Especially, reading the explanation of every line in algorithm 1 was tiring. Maybe this can be shortened. Furthermore, the authors dive head first into the topic of their paper. I suggest to provide an easier introduction for the reader. Shortly explain what anytime algorithms are and how they work.

You state that a lot of previous work is about when to stop, but few work is about tuning parameters during runtime. You should add the work of Gomoluch et al [2] and Speck et al. [1] to the related work. They adapt the hyperparameters of a normal search during execution and they also train neural networks for this. Please relate your works to theirs. One of my biggest concerns is  the novelty of your approach with respect to theirs.

You also stated that Thayer and Ruml presented a method which adapts the weight at runtime. How is this done? Why don't you compare your approach against theirs?

In Figure 4 you have to change how the median values are marked. For many boxplots, I do not see where the median is. Thus, you technique could be better or worse than some static configuration.
Figure 5 confused me more than it informed me. Try to make the content of the subfigure easier to understand.

A final question: Are the time steps and solution qualities in your MDP discrete?

### Minor Issues
- Figure 4: add to each subplot the domain as caption.

### References
1. David Speck, André Biedenkapp, Frank Hutter, Robert Mattmüller and Marius Lindauer.
Learning Heuristic Selection with Dynamic Algorithm Configuration.
In Proceedings of the 31st International Conference on Automated Planning and Scheduling (ICAPS 2021). 2021.
2. Paweł Gomoluch, Dalal Alrajeh, Alessandra Russo, Antonio Bucchiarone.
Learning Neural Search Policies for Classical Planning.
In Proceedings of the 30st International Conference on Automated Planning and Scheduling (ICAPS 2020). 2020.

---

> ### Author Response · Authors · 2021-06-04
> **Author Response**
>
> Thank you for the useful feedback. Your suggestions will help us improve the paper.
>
> We appreciate you pointing out the works of Gomoluch et al. and Speck et al. While our approach was developed independently, we agree that there are similarities that should be mentioned in our related work section. However, since there are important differences, such as our focus on an anytime setting, the works are not directly comparable.
>
> We will try to adapt these approaches for future evaluations. Thank you for suggesting such baselines.
>
> The median is 0 for some of the approaches, which is why the median line is not visible in Figure 4. We will mention this in the caption. Similarly, we will make Figure 5 more clear.
>
> The quality of a solution is inversely proportional to its cost, which is discrete for some benchmark domains, such as the sliding puzzle since we use the number of steps to the goal configuration, but continuous for other benchmark domains, such as the traveling salesman problem since we use the sum of real-valued costs of the edges in the tour. The time steps are discrete in our work but can be continuous. We will clarify this in the paper.
>
> We will address all other minor comments in the paper.

---

### Decision · Program_Chairs · 2021-06-10

**Decision:**

Accept

**Comment:**

Both reviews are positive and the reviewers' questions were well answered in the response, resulting in acceptance.

Nevertheless, the reviewers' questions/concerns should be incorporated in the final version of the paper.